**Data Availability Statement:** Data cannot be shared publicly because of data sharing and

# Associations between recreational cannabis legalization and cannabis-related emergency department visits by age, gender, and geographic status in Ontario, Canada: An interrupted time series study

Chungah Kim[1], Antony Chum[1,2]*, Andrew Nielsen[3], Sara Allin[4], Tarra L. Penney[5,6], Katherine Rittenbach[7,8], Frank P. MacMaster[7,8], Patricia O'Campo[2,4]

1 York University, School of Kinesiology and Health Sciences, Toronto, Ontario, Canada, 2 Unity Health Toronto, MAP Centre for Urban Health Solutions, Toronto, Ontario, Canada, 3 Canadian Institute for Health Information, Toronto, Ontario, Canada, 4 University of Toronto, Dalla Lana School of Public Health, Toronto, Ontario, Canada, 5 York University Faculty of Health, School of Global Health, Toronto, Ontario, Canada, 6 University of Cambridge School of Clinical Medicine, MRC Epidemiology Unit, Cambridge, Cambridgeshire, United Kingdom, 7 Alberta Health Services, Strategic Clinical Networks, Edmonton, Alberta, Canada, 8 University of Calgary, Department of Psychiatry, Calgary, Alberta, Canada

* antonychum@gmail.com

## Abstract

Legalization of recreational cannabis in Ontario included the legalization of flower and herbs (Phase 1, October 2018), and was followed by the deregulation of cannabis retailers and sales of edibles (Phase 2, February 2020). Research on the impact of cannabis legalization on acute care utilization is nascet; no research has investigated potential age, gender, and geographically vulnerable subgroup effects. Residents living in Northern Ontario not only have higher levels of substance use problems, but also have inadequate access to primary healthcare. Our study investigated the impact of Ontario's recreational cannabis policy (including Phase 1 and 2) on cannabis-attributable emergency department (ED) visits, and estimated the impact separately for different age and gender groups, with additional analyses focused on Northern Ontarians. We created a cohort of adults (18 and over) eligible for provincial universal health insurance with continuous coverage from 2015–2021 (n = 14,900,820). An interrupted time series was used to examine the immediate impact and month-to-month changes in cannabis-related ED visits associated with Phase 1 & 2 for each subgroup. While Northern Ontario has higher rates of cannabis-related ED visits, both Northern and Southern Ontario show similar patterns of changes. Phase 1 was associated with significant increases in adults 25–64, with the strongest increases seen in women 45–64. Month-to-month trends were flattened in most groups compared to pre-legalization. Phase 2 was associated with significant immediate increases for adults aged 18–44 in both genders, but the increases were larger in women than men. No significant month-to-month changes were detected in this period. While current preventive efforts are largely focused on reducing cannabis-related harms in youths and younger adults, our results show that adults 25–64, particularly women, have been significantly impacted by cannabis policies.

privacy agreements to which the Institute for Clinical Evaluative Sciences (ICES) are bound. While data sharing agreements prohibit ICES from making the dataset publicly available, access may be granted to those who meet pre-specified criteria for confidential access, available at www.ices.on.ca/DAS. In order to request access to the datasets used in study, please contact ICES DAS at das@ices.on.ca.

**Funding:** Funding for the study was provided by Canadian Institutes of Health Research (Project Grant FRN# 173447, NPI: Antony Chum). AC is also supported by the Canada Research Chair program (CRC-2021-00269). The funding agencies had no role in the design and conduct of the study; the collection, management, analysis or interpretation of the data; the preparation, review or approval of the manuscript; or the decision to submit the manuscript for publication.

**Competing interests:** The authors have declared that no competing interests exist.

Further research on gender-specific cannabis dosage and targeted interventions for adult women should be investigated. Legalization did not appear to have a differential impact on Northern versus Southern Ontario, but higher rates of ED visits in the North should be addressed.

## Introduction

On October 17th, 2018, the federal Canadian government, through the Cannabis Act 2018, legalized the possession, consumption, and sales of cannabis for recreational use for adults. Subsequently, each province implemented their own regulations governing access and use. In Ontario, Canada's most populated province with 38.8% of the entire Canadian population, a two-phased approach was taken for the sales and distribution of cannabis products. During Phase 1 (October 18, 2018-March 17, 2020), flowers and herbs were the only available product and could be exclusively purchased from the government-run website and 67 retail storefronts (licensed through a lottery) [1]. In Phase 2 (March 18, 2020-June 17, 2021), cannabis edibles became legalized and widely available, and the cap on the number of retailers was lifted, and by June 2021, there were 826 stores [2].

Although prior research has shown that cannabis may be associated with fewer acute harm events compared to tobacco, alcohol, and opiates [3], problematic use can lead to anxiety, paranoia, suicidal ideation, cognitive impairment, induce temporary tachycardia and cannabinoid hyperemesis syndrome [4], or lead to cyclical emesis with chronic use [5]. While emerging evidence suggests that legalization has led to modest increases in cannabis-attributable emergency department (ED) visits [6–9], these studies have a number of gaps. First, these studies did not examine a possible differential impact of legalization across age and gender [10]. This information is necessary for the development of age- and gender-specific interventions. Second, there has been no research that looks at the impacts of legalization on cannabis-related ED visits in Northern Ontario. Northern Canada experiences low physician retention [11], a lack of comprehensive service availability, and a historical vulnerability to substance use problems [12]. These issues are likely driven by Northern Ontario's sparsity [13], with only around 800,000 inhabitants across 200,000 km2. This makes the provision of services and infrastructural investments difficult, which in turn exacerbates poverty and unemployment [14].

In Ontario, Northern residents have double the rate of substance-related healthcare encounters compared to the rest of the province (911.6 per 100,000 vs. 457.2 per 100,000) [15], which is in large part attributable to the socioeconomic inequalities and disparities. This underscores the importance of research on the impact of cannabis legalization on this under-served population. Our study will address these distinct gaps in the literature using disaggregated analyses of population-level health administrative data in Ontario.

We aim to investigate the impact of recreational cannabis policy (including Phase 1 and 2) on cannabis-attributable ED visits, and estimate the impact separately for different age and gender groups, with additional analyses focused on Northern Ontarians.

## Materials and methods

### Data source

The study cohort (n = 14,900,820) was created using health-administrative data held at ICES (formerly the Institute for Clinical Evaluative Sciences), based on all Ontarians (contained in

the Registered Person Database) who are eligible for the Ontario Health Insurance Plan (OHIP) as of October 17, 2015, with healthcare records followed through until June 17, 2021. Participants must be 18 or over as of October 17, 2018 (the date of cannabis legalization), and have continuous OHIP coverage and residency in Ontario for the entire study period to be included in the study (October 2015 to May 2021). OHIP is Ontario's universal healthcare that covers over 95% of Ontario residents, those who are not eligible includes individuals in their 3-month waiting periods and migrants with temporary status (e.g. international students, temporary workers) [16], which includes approximately 500,000 people [17]. Age groups for our study were based on each person's age on the day of Canada's cannabis legalization. Northern Ontario, based on Statistics Canada's health regions [18], was defined as the area of the province that is north of the French River and Algonquin Park. The Registered Person Database provides gender identity [19]; however, gender may be mismatched if not officially changed.

This study complied with privacy regulations of the Institute for Clinical and Evaluative Science (ICES). To protect privacy, all cell sizes fewer than six individuals were suppressed and reported as n < 6. No participants were recruited or contacted; our study cohort was built using health administrative data provided by ICES. ICES is an independent, non-profit research institute whose legal status under Ontario's health information privacy law allows it to collect and analyze health care and demographic data used for the study, without consent, for health system evaluation and improvement. All patient information was anonymized and de-identified prior to analysis. Ethics approval for this study was obtained through York University (REB# 2022–254).

## Outcomes

The outcomes of interest were monthly cannabis-attributable ED visits from October 2015 to May 2021 in Ontario (i.e. the frequency of the event within age, gender, and region-specific subgroups for each month of the study period). Using the National Ambulatory Care Reporting System, Discharge Abstract Database, and Ontario Mental Health Reporting System, records were included with at least one of the following ICD-10 codes for either the primary or supplemental diagnosis: F12 (cannabis-related disorders) and T40.7 (cannabis poisoning), following the conventions from prior studies [6, 7, 20]. All hospitals and emergency departments in Ontario report to these databases. DAD, OMERS, and NACRS are updated 4 times per year, and ORGD is updated once per year [21–24].

## Cannabis legalization phases

Our study considers three distinct time periods related to cannabis policy in Ontario, including 1) the pre-legalization period (October 2015-October 2018), 2) Phase 1 of legalization (October 2018-February 2020) which marked the beginning of online flower and herb sales alongside limited physical retail locations, and 3) Phase 2 (March 2020-June 2021) saw the removal of the retail cap and increased edible cannabis availability. These cutoff dates were used in a prior study estimating the impact of legalization and commercialization on cannabis-related ED visits in the Ontario population [9]. Following recommendations from the prior study, healthcare visits in March and April 2020 were censored to account for large decreases in health-seeking behaviors (including ED visits) associated with the early stages of COVID-19 pandemic.

## Statistical analyses

We used an interrupted time-series with single-group, multiple-interventions design to examine the immediate and month-to-month changes associated with cannabis-related policies on

cannabis-related ED visits for each age and gender group in Ontario (both North and South), Southern, and Northern Ontario. We conducted segmented regressions using the negative binomial function to estimate the incidence rate ratio while adjusting for overdispersion (i.e. extra-Poisson variation). We estimated models for each age group (18 to 24, 25 to 44, 45 to 64, and 65+) and gender separately, given prior research finding age and gender differences in the patterns of cannabis use [25]. We adjusted for the length of the month and for any seasonal effect [26]. We conducted two sensitivity analyses: 1) to examine whether the results were robust regardless of the functional form and address possible residual autocorrelation in a time-series dataset, the Prais-Winsten regressions following a first-order autoregressive process were performed [27], and 2) we included the months March and April 2020 in an alternative specification of the models, and used an indicator variable for both months as an adjustment to confirm whether the estimates were robust.

## Results

Table 1 shows the number of individuals with 1 or more cannabis-related ED visits in the entire study period (October 18, 2015-June 17, 2021) across age, gender, and for Northern, Southern, and the entirety of Ontario. In general, men, those in the 18–24 age group, and those living in Northern Ontario are at a higher risk of cannabis-related ED visits. Table 2 shows the gender- and age-specific average monthly rates (per 100,000 people) of cannabis-related ED visits in the pre-legalization period, Phase 1, and Phase 2. Rates for cannabis-related ED visits appear to increase for age groups under 65 after both Phase 1 and Phase 2.

In Southern Ontario (Table 3), there was an increasing trend in cannabis-related ED visits across all gender and age groups during the pre-legalization period: at the lowest end, an increase of 1.6% ED visits per month among women aged 18–24 (IRR = 1.016, 95% CI 1.013–1.020), and at the highest end, an increase of 7.3% ED visits per month among boys aged 0–17 (IRR = 1.073, 95% CI 1.064–1.082). Legalization (Phase 1) was associated with an immediate (level) increase overall, and they were statistically significant in 3 groups: 1) a 20% increase in ED visits among men aged 45–64 (IRR = 1.207, 95% CI 1.015–1.436), 2) a 13% increase among women aged 25–44 (IRR = 1.134, 95% CI 1.000–1.286), and 3) a 29% increase among women

**Table 1. Cohort characteristics and prevalence of cannabis-related ED visits over the study period (n = 14,900,820).**

| | Ontario | | Northern Ontario | | Southern Ontario | |
|---|---|---|---|---|---|---|
| | Number of people in Registered persons dataset | Number of people with 1 or more cannabis-related ED visits (row %) | Number of people in Registered persons dataset | Number of people with 1 or more cannabis-related ED visits (row %) | Number of people in Registered persons dataset | Number of people with 1 or more cannabis-related ED visits (row %) |
| **Women** | | | | | | |
| 18–24 | 611438 | 5066 (0.83%) | 33417 | 366 (1.10%) | 578021 | 4700 (0.81%) |
| 25–44 | 2044802 | 4793 (0.23%) | 101728 | 370 (0.36%) | 1943074 | 4423 (0.23%) |
| 45–64 | 2117535 | 2176 (0.10%) | 123491 | 166 (0.13%) | 1994044 | 2010 (0.10%) |
| 65+ | 1390376 | 545 (0.04%) | 89402 | 39 (0.04%) | 1300974 | 506 (0.04%) |
| **Total** | 7537613 | 15498 (0.21%) | 422449 | 1260 (0.30%) | 7115164 | 14238 (0.20%) |
| **Men** | | | | | | |
| 18–24 | 643746 | 7804 (1.21%) | 35952 | 514 (1.43%) | 607794 | 7290 (1.20%) |
| 25–44 | 2011323 | 9125 (0.45%) | 106586 | 577 (0.54%) | 1904737 | 8548 (0.45%) |
| 45–64 | 2087004 | 2955 (0.14%) | 124550 | 206 (0.17%) | 1962454 | 2749 (0.14%) |
| 65+ | 1173475 | 582 (0.05%) | 80771 | 45 (0.06%) | 1092704 | 537 (0.05%) |
| total | 7363207 | 24075 (0.33%) | 425857 | 1677 (0.40%) | 6937350 | 22398 (0.32%) |

**Table 2.** Average monthly rate of ED visits (per 100,000 individuals) by age and gender in all of Ontario and Northern Ontario across 3 periods: 1) pre-legalization period, 2) Phase 1 and 3) Phase 3.

| | Ontario | | | | Northern Ontario | | | | Southern Ontario | | | |
|---|---|---|---|---|---|---|---|---|---|---|---|---|
| | Longer pre-legalization period (SD)[1] | Shorter pre-legalization period (SD)[2] | Phase 1 (SD)[3] | Phase 2 (SD)[4] | Longer pre-legalization period (SD)[1] | Shorter pre-legalization period (SD)[2] | Phase 1 (SD)[3] | Phase 2 (SD)[4] | Longer pre-legalization period (SD)[1] | Shorter pre-legalization period (SD)[2] | Phase 1 (SD)[3] | Phase 2 (SD)[4] |
| **Women** | | | | | | | | | | | | |
| 18–24 | 16.94 (3.50) | 19.77 (2.92) | 24.09 (2.41) | 31.07 (3.98) | 23.86 (8.33) | 28.00 (8.83) | 35.73 (17.67) | 56.17 (19.44) | 16.54 (3.47) | 19.29 (2.94) | 23.42 (2.46) | 29.62 (3.67) |
| 25–44 | 5.64 (0.94) | 5.64 (0.94) | 7.19 (0.54) | 9.53 (1.53) | 7.21 (2.98) | 8.99 (2.67) | 12.61 (3.37) | 22.68 (5.30) | 4.49 (1.14) | 5.46 (0.92) | 6.90 (0.51) | 8.84 (1.46) |
| 45–64 | 1.98 (0.55) | 1.98 (0.55) | 2.75 (0.51) | 2.93 (0.39) | 1.84 (1.36) | 2.26 (1.63) | 4.24 (2.21) | 4.17 (2.16) | 1.56 (0.55) | 1.97 (0.54) | 2.66 (0.51) | 2.85 (0.36) |
| 65+ | 0.69 (0.21) | 0.69 (0.21) | 0.93 (0.27) | 0.92 (0.24) | 0.56 (0.91) | 0.88 (1.18) | 1.38 (1.40) | 0.60 (0.74) | 0.50 (0.24) | 0.68 (0.23) | 0.90 (0.26) | 0.94 (0.28) |
| **Men** | | | | | | | | | | | | |
| 18–24 | 26.47 (5.22) | 31.00 (2.86) | 33.89 (2.75) | 37.26 (3.70) | 32.06 (8.21) | 35.76 (8.01) | 48.59 (13.91) | 52.42 (6.31) | 26.14 (5.40) | 30.72 (3.10) | 33.02 (3.03) | 36.36 (3.82) |
| 25–44 | 10.10 (2.05) | 11.81 (1.37) | 13.89 (1.30) | 16.46 (1.61) | 12.14 (3.68) | 14.61 (2.98) | 20.92 (5.60) | 28.44 (8.03) | 9.99 (2.06) | 11.66 (1.45) | 13.50 (1.35) | 15.79 (1.56) |
| 45–64 | 2.35 (0.64) | 2.96 (0.39) | 4.15 (0.71) | 4.63 (0.57) | 2.79 (1.68) | 3.38 (1.84) | 5.53 (2.02) | 8.96 (1.73) | 2.33 (0.63) | 2.93 (0.38) | 4.06 (0.74) | 4.35 (0.63) |
| 65+ | 0.67 (0.35) | 0.84 (0.27) | 1.30 (0.41) | 1.12 (0.32) | 0.48 (0.99) | 0.80 (1.25) | 1.89 (1.64) | 1.62 (1.17) | 0.68 (0.37) | 0.84 (0.33) | 1.26 (0.43) | 1.08 (0.35) |

[1] Oct 2015-Oct 2018.

[2] Aug 2017-Oct 2018. Given the relatively large change in rates over the pre-legalization period, a shorter (14 months) pre-legalization period is given so it is more comparable to Phase 1 and 2.

[3] Oct 2018-Feb 2020

[4] May 2020-May 2021

aged 45–64 (IRR = 1.289, 95% CI 1.020–1.628). In Phase 1, there were no significant month-to-month (trend) increases in ED visits across all gender and age groups.

In Southern Ontario during Phase 2, immediate (level) increases in ED visits were observed among 4 groups: 1) a 36% increase among men aged 18–24 (IRR: 1.363, 95% CI 1.193–1.557), 2) a 26% increase in men aged 25–44 (IRR: 1.222, 95% CI 1.074–1.472), 3) a 50% increase in women aged 18–24 (IRR: 1.499, 95% CI 1.252–1.794), and 4) a 42% increase in women aged 25–44 (IRR: 1.421, 95% CI 1.196–1.688). Finally, in Phase 2, month-to-month (trend)

**Table 3.** Associations between cannabis policies and ED visits in <u>Southern Ontario</u> using <u>negative binomial regression</u> with <u>censored months</u> in the early stages of COVID-19 pandemic.

| | Men | | | | Women | | | |
|---|---|---|---|---|---|---|---|---|
| | 18–24 | 25–44 | 45–64 | 65+ | 18–24 | 25–44 | 45–64 | 65+ |
| Pre-intervention trend | **1.018***<br>**(1.015 to 1.021)** | **1.017***<br>**(1.014 to 1.020)** | **1.024***<br>**(1.019 to 1.029)** | **1.028***<br>**(1.014 to 1.043)** | **1.016***<br>**(1.013 to 1.020)** | **1.020***<br>**(1.016 to 1.024)** | **1.022***<br>**(1.016 to 1.029)** | **1.035***<br>**(1.024 to 1.045)** |
| Legalization level change | 1.025<br>(0.930 to 1.130) | 1.061<br>(0.946 to 1.191) | **1.207***<br>**(1.015 to 1.436)** | 1.401<br>(0.844 to 2.328) | 1.135<br>(0.995 to 1.295) | **1.134***<br>**(1.000 to 1.286)** | **1.289***<br>**(1.020 to 1.628)** | 1.200<br>(0.837 to 1.720) |
| Legalization trend | **0.987****<br>**(0.979 to 0.996)** | 0.994<br>(0.985 to 1.004) | 0.996<br>(0.981 to 1.010) | 0.975<br>(0.932 to 1.018) | 0.992<br>(0.981 to 1.003) | 0.997<br>(0.987 to 1.008) | 0.991<br>(0.971 to 1.011) | 0.982<br>(0.952 to 1.013) |
| Edibles/COVID level change | **1.363***<br>**(1.193 to 1.557)** | **1.258****<br>**(1.074 to 1.472)** | 1.112<br>(0.876 to 1.412) | 1.351<br>(0.668 to 2.732) | **1.499***<br>**(1.252 to 1.794)** | **1.421***<br>**(1.196 to 1.688)** | 1.179<br>(0.853 to 1.630) | 1.416<br>(0.858 to 2.336) |
| Edibles/COVID trend | **0.986***<br>**(0.973 to 0.999)** | 0.996<br>(0.981 to 1.011) | 1.001<br>(0.978 to 0.987) | 0.969<br>(0.899 to 1.039) | 0.984<br>(0.967 to 1.002) | **0.982***<br>**(0.966 to 0.999)** | 0.997<br>(0.966 to 1.029) | 0.978<br>(0.928 to 1.028) |
| Difference pre-intervention vs. legalization trend | **0.970***<br>**(0.962 to 0.978)** | **0.978***<br>**(0.968 to 0.988)** | **0.972***<br>**(0.957 to 0.987)** | **0.948***<br>**(0.906 to 0.992)** | **0.976***<br>**(0.965 to 0.988)** | **0.978***<br>**(0.967 to 0.989)** | **0.969****<br>**(0.950 to 0.990)** | **0.950****<br>**(0.920 to 0.981)** |
| Difference legalization vs. edibles/COVID trend | 0.998<br>(0.983 to 1.014) | 1.002<br>(0.984 to 1.020) | 1.005<br>(0.978 to 1.033) | 0.995<br>(0.915 to 1.081) | 0.992<br>(0.972 to 1.013) | 0.985<br>(0.966 to 1.005) | 1.007<br>(0.969 to 1.045) | 0.996<br>(0.939 to 1.057) |

**Table 4. Associations between cannabis policies and ED visits in Northern Ontario using negative binomial regression with censored months in the early stages of COVID-19 pandemic.**

| | Men | | | | Women | | | |
|---|---|---|---|---|---|---|---|---|
| | 18–24 | 25–44 | 45–64 | 65+ | 18–24 | 25–44 | 45–64 | 65+ |
| Pre-intervention trend | 1.008 (0.999 to 1.016) | **1.014**[**] **(1.005 to 1.023)** | **1.027**[**] **(1.010 to 1.045)** | **1.109**[**] **(1.033 to 1.190)** | **1.016**[**] **(1.005 to 1.027)** | **1.024**[***] **(1.013 to 1.034)** | 1.015 (0.994 to 1.035) | **1.053**[*] **(1.001 to 1.107)** |
| Legalization level change | 1.021 (0.758 to 1.375) | 1.108 (0.791 to 1.468) | 1.144 (0.649 to 2.015) | 1.378 (0.330 to 5.740) | 1.031 (0.704 to 1.509) | 1.212 (0.851 to 1.727) | **2.004**[*] **(1.017 to 3.949)** | 1.683 (0.448 to 6.324) |
| Legalization trend | **1.031**[*] **(1.005 to 1.057)** | **1.022**[*] **(1.000 to 1.044)** | 1.013 (0.968 to 1.058) | 0.965 (0.864 to 1.065) | 1.008 (0.977 to 1.039) | 1.016 (0.995 to 1.038) | 0.986 (0.931 to 1.041) | 0.949 (0.841 to 1.056) |
| Edibles/COVID level change | 0.762 (0.510 to 1.139) | 0.917 (0.604 to 1.391) | 1.402 (0.663 to 2.966) | 0.772 (0.132 to 4.504) | 1.644 (0.983 to 2.745) | **1.747**[*] **(1.084 to 2.814)** | 0.720 (0.285 to 1.822) | 0.416 (0.052 to 0.329) |
| Edibles/COVID trend | 1.003 (0.966 to 1.040) | 1.020 (0.982 to 1.058) | 0.998 (0.923 to 1.072) | 1.065 (0.884 to 1.247) | 0.981 (0.932 to 1.030) | 0.998 (0.967 1.029) | 1.076 (0.980 to 1.171) | 1.086 (0.882 to 1.291) |
| Difference pre-intervention vs. legalization trend | 1.024 (0.996 to 1.052) | 1.012 (0.985 to 1.040) | 0.986 (0.940 to 1.034) | **0.870**[*] **(0.770 to 0.983)** | 0.992 (0.960 to 1.025) | 0.975 (0.946 to 1.006) | 0.972 (0.917 to 1.030) | 0.901 (0.800 to 1.015) |
| Difference legalization vs. edibles/COVID trend | 0.973 (0.929 to 1.018) | 0.993 (0.947 to 1.041) | 0.985 (0.902 to 1.075) | 1.106 (0.893 to 1.369) | 0.973 (0.917 to 1.031) | 1.002 (0.948 to 1.059) | 1.094 (0.978 to 1.222) | 1.148 (0.908 to 1.451) |

increases had stabilized in all age and gender groups, with a decreasing monthly trend of 1.3% per month among men aged 18–24 (IRR: 0.986, 95% CI 0.973–0.999).

The pre-legalization trend and immediate effect of Phase 1 observed in Northern Ontario (Table 4) were similar in magnitude compared to Southern Ontario. For instance, almost all age and gender groups had significant month-to-month increases in ED visits in the pre-legalization period, and there were immediate insignificant increases for most demographic groups, but only significant among women aged 45–64 (IRR: 2.004, 95% CI 1.017–3.949). In Phase 2, while many demographic groups had immediate increases in ED visits in Southern Ontario, fewer groups observed immediate increases in Northern Ontario (only women aged 25–44 had a statistically significant increase of 75%, IRR 1.747, 95% CI 1.084–2.814). In Phase 2 (i.e. the edibles period), similar to Southern Ontario, the month-to-month increase in ED visits also stabilized in Northern Ontario. Table 5 provides the estimates for the entirety of Ontario (i.e. pooling Northern and Southern Ontario): these results are substantially similar to Southern Ontario given that its residents account for 94.3% of the entire province's population. Fig 1 visualizes the change in monthly cannabis-related ED visits by each subgroup.

## Discussion

Prior to cannabis legalization, we observed an increasing trend in cannabis-related ED visits in all subpopulation groups, which may reflect decreased stigma associated with recreational cannabis-use over time [28]. Cannabis legalization was associated with immediate increases in ED visits of adults in Ontario. Phase 1 was associated with significant immediate increases in adults 25–64, and Phase 2 was associated with immediate increases in those 18–44, with greater increases observed in women for both phases. Among these groups, the pattern is characterized by rapid immediate increase, but no significant subsequent trend (month-to-month) change is observed. Also, while Northern Ontario has higher rates of cannabis-related ED visits, both Northern and Southern Ontario show relatively similar patterns of changes with regards to Phase 1 and 2 of legalization.

**Table 5. Associations between cannabis policies and <u>ED visits</u> in all of <u>Ontario</u> using <u>negative binomial regression</u> with <u>censored months</u> in the early stages of COVID-19 pandemic.**

| | Men | | | | Women | | | |
|---|---|---|---|---|---|---|---|---|
| | 18–24 | 25–44 | 45–64 | 65+ | 18–24 | 25–44 | 45–64 | 65+ |
| Pre-legalization trend | **1.017***<br>**(1.014 to 1.019)** | **1.017***<br>**(1.013 to 1.020)** | **1.024***<br>**(1.019 to 1.029)** | **1.031***<br>**(1.017 to 1.046)** | **1.016***<br>**(1.012 to 1.020)** | **1.020***<br>**(1.017 to 1.024)** | **1.021***<br>**(1.015 to 1.028)** | **1.036***<br>**(1.026 to 1.046)** |
| Legalization level change | 1.025<br>(0.934 to 1.125) | 1.059<br>(0.945 to 1.187) | **1.199***<br>**(1.015 to 1.417)** | 1.432<br>(0.885 to 2.315) | 1.128<br>(0.990 to 1.285) | **1.140***<br>**(1.011 to 1.286)** | **1.339***<br>**(1.059 to 1.692)** | 1.241<br>(0.889 to 1.733) |
| Legalization trend | **0.991***<br>**(0.983 to 0.998)** | 0.997<br>(0.987 to 1.007) | 0.997<br>(0.983 to 1.011) | 0.974<br>(0.934 to 0.985) | 0.994<br>(0.983 to 1.004) | 0.998<br>(0.988 to 1.008) | 0.991<br>(0.971 to 1.010) | 0.979<br>(0.951 to 1.008) |
| Edibles/COVID level change | **1.303***<br>**(1.148 to 1.480)** | **1.222***<br>**(1.046 to 1.427)** | 1.142<br>(0.908 to 1.437) | 1.304<br>(0.671 to 2.535) | **1.515***<br>**(1.268 to 1.811)** | **1.447***<br>**(1.227 to 1.707)** | 1.135<br>(0.820 to 1.571) | 1.327<br>(0.832 to 2.115) |
| Edibles/COVID trend | **0.987***<br>**(0.975 to 0.999)** | 0.998<br>(0.984 to 1.013) | 1.000<br>(0.978 to 1.023) | 0.977<br>(0.910 to 1.043) | 0.984<br>(0.967 to 1.001) | 0.985<br>(0.969 to 1.001) | 1.003<br>(0.971 to 1.034) | 0.985<br>(0.938 to 1.032) |
| Difference pre-legalization vs. legalization trend | **0.973***<br>**(0.965 to 0.981)** | **0.981***<br>**(0.971 to 0.990)** | **0.973***<br>**(0.959 to 0.988)** | **0.945****<br>**(0.906 to 0.986)** | **0.977***<br>**(0.966 to 0.988)** | **0.978***<br>**(0.967 to 0.988)** | **0.970****<br>**(0.950 to 0.990)** | **0.945***<br>**(0.917 to 0.974)** |
| Difference legalization vs. edibles/COVID trend | 0.996<br>(0.982 to 1.011) | 1.001<br>(0.984 to 1.019) | 1.003<br>(0.977 to 1.030) | 1.002<br>(0.927 to 1.190) | 0.990<br>(0.970 to 1.010) | 0.987<br>(0.969 to 1.006) | 1.012<br>(0.974 to 1.051) | 1.006<br>(0.952 to 1.063) |

Our results are largely consistent with findings from prior research that found cannabis legalization (Phase 1) [6] and cannabis edibles legalization and commercialization (Phase 2) were associated with increases in cannabis-related ED visits in Canadian provinces [8, 9]. For example, a study reported that cannabis legalization (Phase 1) led to a small immediate increase in urban-Alberta cannabis-related ED visits; however, this study included a relatively short period (less than 1 year) and did not adjust for temporal autocorrelation and seasonality [6]. Our results also support the findings of an Ontario study which suggest that the immediate effects of cannabis legalization (Phase 1) were largely driven by adults over the age of 24 [9]. Additionally, similar to other studies that examined the Phase 2 effect (i.e. edibles and commercialization), we found evidence of an immediate increase in cannabis-related ED visits. Only one study from Hamilton, Ontario contradicted our results: they found that there was no change in the rate of cannabis-related ED visits following legalization [29], but the authors only examined a single hospital, and did not measure level and trend changes.

There are some potential explanations why the initial legalization (i.e. phase 1) affected older adults (25–64), how commercialization/edibles coincided with increases in younger adults (18–44) ED visits, and why women experienced higher increases. First, following legalization in Canada, research has shown that older adults were more likely to try cannabis for the first time (compared to pre-legalization) because of destigmatization and the changing legal landscape [30], and in turn, these older inexperienced users may be more likely to overdose and would require ED visits [31]. Second, there is research showing that younger users prefer edibles to smoking because of its appearance (i.e. more colorful) and taste (i.e. integrated into candies and baked goods) [32, 33]; therefore, it may not be surprising that the introduction of edibles coincided with increased incidents among younger adults. Lastly, in a study using out-patient data from the US, while men report higher rates of cannabis-use disorder than women, women appear to have a faster trajectory from cannabis first-use to developing cannabis-use disorder [34], and the faster trajectory may be associated with increased acute care utilization.

It is important to put these seemingly large relative increases into context. The issues associated with Northern Ontario (e.g. higher substance use problems) seems to be reflected in consistently higher rates of ED visits in each subpopulation group in the North; however, the

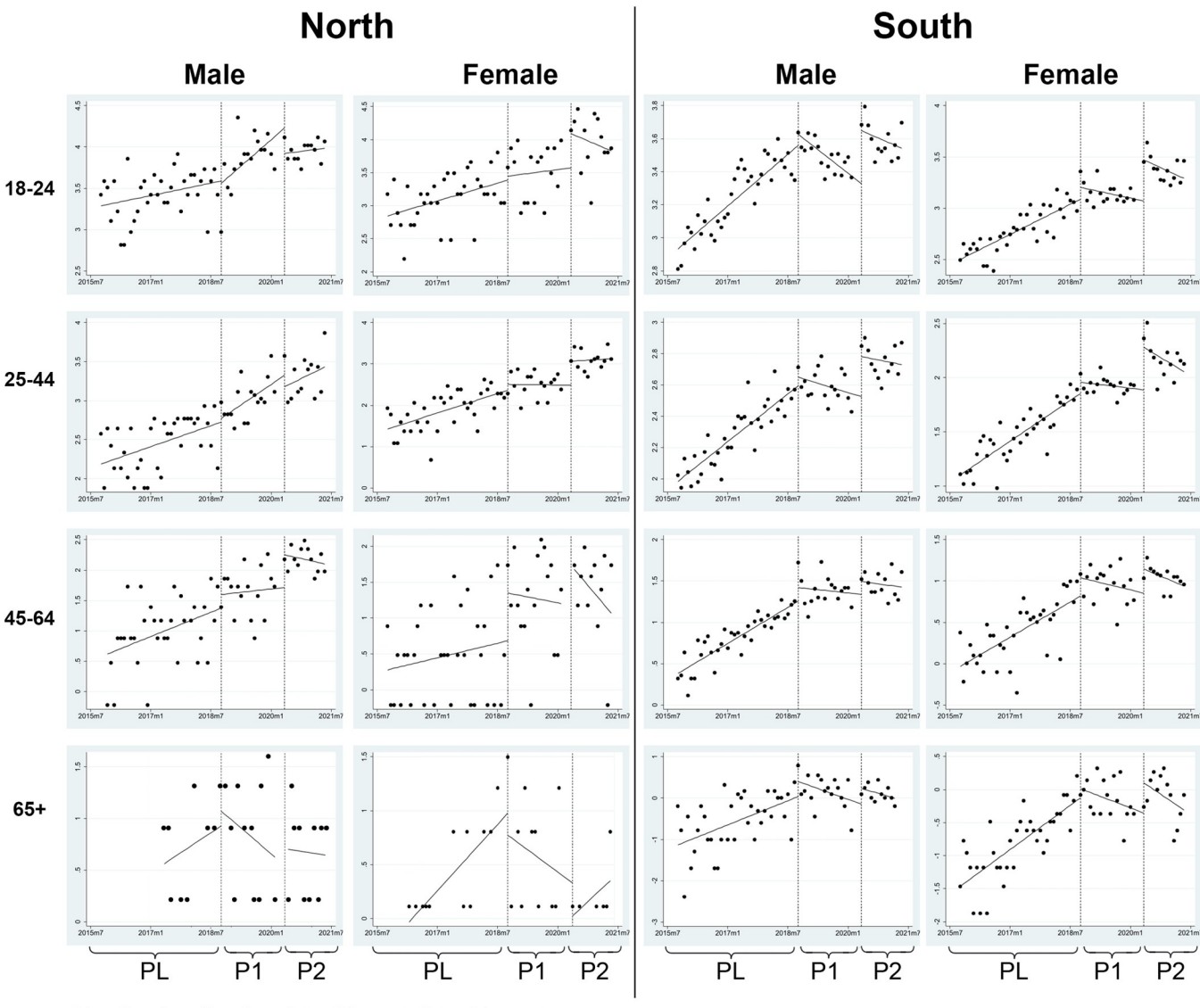

**Fig 1. Change in monthly cannabis-related emergency department visits by gender, age, and geography (i.e. Northern Ontario vs Southern Ontario).**

effects of legalization appear to be uniform across northern and southern Ontario. A possible explanation is that Northern Ontario may be protected by the relatively lower density of cannabis retailers [35]. However, further research is necessary to investigate the association between retailer availability and cannabis-related ED visits.

While our study cannot disentangle the effects of COVID vs. commercialization, we can better contextualize our results by looking at changes in cannabis sales over the COVID period. There is evidence that the COVID-19 pandemic was associated with a 25% increase in monthly cannabis sales in Canada, relative to the counterfactual (constructed from the pre-pandemic trend) [36]. More specifically, the pandemic-related increased sales found in the prior study was driven by cannabis retailers, which may explain the increased number of sub-population groups that had significantly increases in ED visits, particularly among younger adults in Southern Ontario (given that it has higher retailer density compared to the North).

Our study is the first to conduct age- and gender-specific analyses on legalization and cannabis-related ED visits in Canada. Since prior studies did not include age and gender stratifications, they were unable to identify whether the effects were consistent across age and gender groups, and whether the apparent effects may be driven by a specific subgroup. Our study also makes unique contributions to the study of Northern Ontario. While our study finds consistently higher rates of cannabis-related ED visits in Northern Ontario compared to the rest of the province (consistent with studies of substance-related healthcare visits in the North prior to cannabis legalization [15]), we note that ED visits in the North and South were similarly affected by Phase 1 and Phase 2.

Limitations of this study include: 1) the use of single-group interrupted time-series is vulnerable to concurrent interventions, policies or events that may confound the relationship between cannabis legalization and cannabis-related ED visits. While we have adjusted for the early COVID-19 effects on ED visits, there may be other unforeseen events that impact people's willingness to visit the ED. 2) Phase 2 of the study overlapped with the COVID-19 pandemic, so we cannot confidently estimate the independent effects associated with commercialization and edibles. 3) Our study relies on administrative health records to indicate whether individuals have continuous OHIP coverage and residency in Ontario, but some individuals may not be in Ontario for the full study period (e.g. traveling while still maintaining OHIP coverage). 4) The reporting of cannabis-related ED visits may have changed over time, especially given that patients may feel less stigmatized reporting cannabis use at the ED after legalization, which may bias our results towards a significant finding for Phase 1 effects; however, prior literature on cannabis-related ED visits uses the same methods for identifying outcomes [37].

Future studies could take a different approach to isolate the edibles/commercialization effects on cannabis-related ED visits. Such a study can include a control region that is contextually similar (e.g. comparison with a US state that was similarly impacted by COVID-19, but did not implement cannabis legalization) so that we can isolate the Phase 2 effects. Further research should be conducted on the impact of cannabis legalization on the cannabis-related ED visits and hospitalizations in specific patient and at-risk groups (e.g. people with depression, anxiety, or schizophrenia).

## Conclusion

Our study provides evidence that the rate of increase in cannabis-related ED visits in Ontario has flattened even in the face of cannabis legalization and deregulation of retailers/edibles. Furthermore, the association between cannabis legalization and cannabis-related ED visits varies across age, gender, and geography (i.e., Northern vs Southern Ontario). Of note, cannabis-related ED visits in adults aged 18–44 in both genders in the South and women aged 25–44 in the North were significantly increased during Phase 2; however, it is unknown how much COVID-19 contributed to these changes. While legalization (Phase 1 and 2) did not differentially impact the North (vs the South), cannabis-related ED visits remain high and should be addressed in this region. While current preventive efforts are largely focused on reducing cannabis-related harms in youths and in school settings [38], our results show that adults 25–64, particularly women, have been significantly impacted by cannabis policies. Further research on gender-specific cannabis dosage and targeted interventions for adults should be investigated. Legalization did not appear to have a differential impact on Northern versus Southern Ontario, but higher rates of ED visits in the North should be addressed.

## Author Contributions

**Conceptualization:** Antony Chum.

**Data curation:** Antony Chum, Andrew Nielsen.

**Formal analysis:** Chungah Kim, Antony Chum, Sara Allin.

**Methodology:** Chungah Kim.

**Writing – original draft:** Chungah Kim, Antony Chum.

**Writing – review & editing:** Chungah Kim, Antony Chum, Andrew Nielsen, Sara Allin, Tarra L. Penney, Katherine Rittenbach, Frank P. MacMaster, Patricia O'Campo.

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
