## [Decision Letter · Decision Letter 0]

3 Aug 2022

PONE-D-22-11906Associations between recreational cannabis legalization and cannabis-related emergency department visits by age, gender, and geographic status in Ontario, Canada: an interrupted time series studyPLOS ONE

Dear Dr. Chum,

Thank you for submitting your manuscript to PLOS ONE. After careful consideration, we feel that it has merit but does not fully meet PLOS ONE’s publication criteria as it currently stands. Therefore, we invite you to submit a revised version of the manuscript that addresses the points raised during the review process.

We look forward to receiving your revised manuscript.

Kind regards,

Raphael Mendonça Guimaraes, PhD

Academic Editor

PLOS ONE

Journal Requirements:

Reviewers' comments:

Reviewer's Responses to Questions

**Comments to the Author**

1. Is the manuscript technically sound, and do the data support the conclusions?

Reviewer #1: Yes

Reviewer #2: Yes

2. Has the statistical analysis been performed appropriately and rigorously? 

Reviewer #1: Yes

Reviewer #2: Yes

3. Have the authors made all data underlying the findings in their manuscript fully available?

Reviewer #1: No

Reviewer #2: Yes

4. Is the manuscript presented in an intelligible fashion and written in standard English?

Reviewer #1: Yes

Reviewer #2: Yes

5. Review Comments to the Author

Reviewer #1: I congratulate the authors for this properly designed and written study that addresses a topic dear to public health.

I point out minor revision that, in my opinion, can improve the discussion section of the study.

# although the comparisons of the results of this study with the results of others are adequately made, I believe it is necessary to point out possible explanations for the differences in gender and age found. In the discussion section, try to answer the following question: "why have adults aged 25 to 64, particularly women, been significantly impacted by cannabis policies?"

Reviewer #2: The sentence as follows is confusing:

“These studies focus only on the GENERAL POPULATION or the pediatric population, which ignores the differential impact of legalization across gender and lifecourse (10).” (verbatim, emphasis added)

The limitations do NO have any association with the fact one is assessing the general population. The potential limitations are putatively associated with THE WAY the general population may be assessed (Census data? Probability samples? Convenience samples?) and the way the general population is STRATIFIED.

There is no problem to stratify the general population by gender, age, ethnicity, etc… and lifecourse is perfectly compatible with different sampling strategies or census data. The issue here refers to problems secondary to statistical inference for non-probability samples (see, for instance: https://projecteuclid.org/journals/statistical-science/volume-32/issue-2/Inference-for-Nonprobability-Samples/10.1214/16-STS598.full) and/or the absence of any stratification (or “en bloc” analyses).

Please, clarify the sentence as follows for non-Canadian readers (remembering PLOS One has a large audience, worldwide):

“Northern Canada experiences low physician retention (11), a lack of comprehensive service availability, and a historical vulnerability to substance use problems (12)”

I do believe the authors are describing sound facts and the references document such facts. But something has made Northern Canada a cluster of many different problems. Why?

Maybe the population is too sparse, so there are few services. Another putative reason may be the relative poverty of the region vis-à-vis affluent areas. Maybe there are few and/or low paid jobs, maybe there is an absence of leisure activities, maybe bad weather makes the lives of people a hell, maybe people feel such region is a boring place… The set of “maybe” hypothesis is open.

These (or other ones?) hypotheses should be mentioned by the authors (here or in the Discussion). They have first-hand information to be shared with potential readers.

One famous Canadian said there is a Globalization of Addiction (https://www.amazon.com/Globalization-Addiction-Study-Poverty-Spirit/dp/0199588716/ref=sr_1_1?crid=2IATPE7N9PVH5&keywords=the+globalization+of+addiction&qid=1659516078&s=books&sprefix=the+globalization+of+addiction%2Cstripbooks%2C189&sr=1-1) and used the expression: “A study in poverty of spirit”. Something is taking place in Northern Canada, adding a local layer to the global dimension of addictions. Please, provide a brief explanation!

We have here too much demand, not enough supply, i.e. the classic definition of a far from optimal supply-demand curve. Why?

Please provide some information, maybe figures about the comprehensiveness of the OHIP. The text mentions:

Based on all Ontarians (contained in the Registered Person Database) eligible for the Ontario Health Insurance Plan (OHIP).

The key questions are as follows: WHO are NOT eligible? Such non-eligible fraction could be defined as Tiny?, Large?, Easy to understand (for instance, excluding illegal immigrants)? Far from optimal, but based on fair norms? Unfair?

Basic information about the size and composition of the set of people defined as non-eligible will help the potential readers to understand whether this putative bias is relevant or not.

Please, clarify whether the Ontario Mental Health Reporting System is comprehensive and has been regularly updated over time.

What is the role of the private sector? Negligible? Should people who are NOT covered by the OHIP pay their medical expenses “out-of-pocket” or are reimbursed by private insurance companies (of course in case there are such companies, as in the US health system).

The “pre-legalization period (October 2015-October 2018)” (verbatim) cannot be described under the label “Intervention”. Maybe this section could be rather called “Baseline data AND intervention”. There is no intervention from October 2015-October 2018, just more of the same, i.e. the previous status quo.

The statistical analyses are sound. The missing point is to state: we performed two sensitivity analyses and they showed that… WHAT? (e.g. we found “the sufficient convergence of the AR(1) coefficient [was] reached”, quoting https://cran.r-project.org/web/packages/prais/prais.pdf).

There is a kind of ecological (?) trend towards an increase of cannabis use over time, even during the period NO intervention has been in place:

“In Southern Ontario (Table 3), there was an increasing trend in cannabis-related ED visits across all gender and age groups during the pre-legalization period (verbatim)”

The influence of the actual interventions is clear, but pre-intervention trends should be better discussed. Maybe there is no definitive explanation, but much probably there are some hypotheses: changing attitudes and mores?, a higher use among younger birth cohorts, even in the absence of any legal change?, etc.

Please, explain how a place described by the authors themselves (in the Introduction) as SO different from other regions shows trends roughly comparable to other Canadian provinces:

“Our results are largely consistent with findings from prior research that found

cannabis legalization (Phase 1) (6) and cannabis edibles legalization and commercialization

(Phase 2) were associated with increases in cannabis-related ED visits in Canadian provinces”

The COVID pandemic should not be addressed as nothing else but a limitation.

“Phase 2 of the study overlapped with the COVID-19 pandemic, so we cannot confidently estimate the independent effects associated with commercialization and edibles.”

Of course, it is!, the authors are 100% right!

But what happened in the region under analysis during this period?

Several papers, worldwide show an increase (whereas other studies have shown a decrease or no major change) in the use of different substances in consequence of higher levels of anxiety (among other factors). See, for instance, a recent paper from Canada (https://substanceabusepolicy.biomedcentral.com/articles/10.1186/s13011-022-00441-x).

Besides being a limitation in the sense of confusing the findings of the intervention, what do the authors know or hypothesize about what has happened in the region under analysis? As described by the authors themselves, this Region seems to be quite different from other contexts.

6. PLOS authors have the option to publish the peer review history of their article (what does this mean?). If published, this will include your full peer review and any attached files.

Reviewer #1: **Yes: **Rafael Tavares Jomar

Reviewer #2: **Yes: **Francisco I. Bastos

---

## [Author Response · Author response to Decision Letter 0]

21 Sep 2022

Reviewer #1: 

I congratulate the authors for this properly designed and written study that addresses a topic dear to public health.

1. I point out minor revision that, in my opinion, can improve the discussion section of the study. Although the comparisons of the results of this study with the results of others are adequately made, I believe it is necessary to point out possible explanations for the differences in gender and age found. In the discussion section, try to answer the following question: "why have adults aged 25 to 64, particularly women, been significantly impacted by cannabis policies?"

Thank you for your comment. We have now improved the discussion section by providing extended explanations for the differences in gender and age. It now reads:

“There are some potential explanations why the initial legalisation (i.e. phase 1) affected older adults (25-64), how commercialization/edibles coincided with increases in younger adults (18-44) ED visits, and why women experienced higher increases. First, following legalisation in Canada, research has shown that older adults were more likely to try cannabis for the first time (compared to pre-legalisation) because of destigmatization and the changing legal landscape (cite), and in turn, these older inexperienced users may be more likely to overdose and would require ED visits (cite). Second, there is research showing that younger users prefer edibles to smoking because of its appearance (i.e. more colourful) and taste (i.e. integrated into candies and baked goods) (cite); therefore, it may not be surprising that the introduction of edibles coincided with increased incidents among younger adults. Lastly, in a study using outpatient data from the US, while men report higher rates of cannabis-use disorder than women, women appear to have a faster trajectory from cannabis first-use to developing cannabis-use disorder (cite), and the faster trajectory may be associated with increased acute care utilisation” (p.10) 

Reviewer #2: 

2. The sentence as follows is confusing: “These studies focus only on the GENERAL POPULATION or the pediatric population, which ignores the differential impact of legalization across gender and lifecourse (10).” (verbatim, emphasis added)

The limitations do NOT have any association with the fact one is assessing the general population. The potential limitations are putatively associated with THE WAY the general population may be assessed (Census data? Probability samples? Convenience samples?) and the way the general population is STRATIFIED. There is no problem to stratify the general population by gender, age, ethnicity, etc… and lifecourse is perfectly compatible with different sampling strategies or census data. The issue here refers to problems secondary to statistical inference for non-probability samples (see, for instance: https://projecteuclid.org/journals/statistical-science/volume-32/issue-2/Inference-for-Nonprobability-Samples/10.1214/16-STS598.full) and/or the absence of any stratification (or “en bloc” analyses).

Thank you for your comment. Prior studies also used complete health administrative data that contained all acute care records - and sampling was not an issue. The word limitation has now been changed to “gap”. We have rephrased the sentence and it now reads: 

“While emerging evidence suggests that legalization has led to modest increases in cannabis-attributable emergency department (ED) visits(6–9), these studies have a number of gaps. First, these studies did not examine a possible differential impact of legalization across age and gender (10).” (p.3)

3. Please, clarify the sentence as follows for non-Canadian readers (remembering PLOS One has a large audience, worldwide): “Northern Canada experiences low physician retention (11), a lack of comprehensive service availability, and a historical vulnerability to substance use problems (12)” I do believe the authors are describing sound facts and the references document such facts. But something has made Northern Canada a cluster of many different problems. Why? Maybe the population is too sparse, so there are few services. Another putative reason may be the relative poverty of the region vis-à-vis affluent areas. Maybe there are few and/or low paid jobs, maybe there is an absence of leisure activities, maybe bad weather makes the lives of people a hell, maybe people feel such region is a boring place… The set of “maybe” hypothesis is open. These (or other ones?) hypotheses should be mentioned by the authors (here or in the Discussion). They have first-hand information to be shared with potential readers. One famous Canadian said there is a Globalization of Addiction (https://www.amazon.com/Globalization-Addiction-Study-Poverty-Spirit/dp/0199588716/ref=sr_1_1?crid=2IATPE7N9PVH5&keywords=the+globalization+of+addiction&qid=1659516078&s=books&sprefix=the+globalization+of+addiction%2Cstripbooks%2C189&sr=1-1) and used the expression: “A study in poverty of spirit”. Something is taking place in Northern Canada, adding a local layer to the global dimension of addictions. Please, provide a brief explanation! We have here too much demand, not enough supply, i.e. the classic definition of a far from optimal supply-demand curve. Why?

We added a sentence for the reason be behind the vulnerability faced by Northern Ontario residents:

“These issues are likely driven by Northern Ontario’s sparsity(13), with only around 800,000 inhabitants across 200,000 km2. This makes the provision of services and infrastructural investments difficult, which in turn exacerbates poverty and unemployment (14).” (p.4)

4. Please provide some information, maybe figures about the comprehensiveness of the OHIP. The text mentions: Based on all Ontarians (contained in the Registered Person Database) eligible for the Ontario Health Insurance Plan (OHIP).The key questions are as follows: WHO are NOT eligible? Such non-eligible fraction could be defined as Tiny?, Large?, Easy to understand (for instance, excluding illegal immigrants)? Far from optimal, but based on fair norms? Unfair? Basic information about the size and composition of the set of people defined as non-eligible will help the potential readers to understand whether this putative bias is relevant or not.

We added a more detailed explanation about the OHIP to help readers better understand the context. It now reads:

“Participants must be 18 or over as of October 17, 2018 (the date of cannabis legalization), and have continuous OHIP coverage and residency in Ontario for the entire study period to be included in the study (October 2015 to May 2021). OHIP is Ontario’s universal healthcare that covers over 95% of Ontario residents, those who are not eligible includes individuals in their 3-month waiting periods and migrants with temporary status (e.g. international students, temporary workers)(16), which includes approximately 500,000 people (17).” (p.4-5)

5. Please, clarify whether the Ontario Mental Health Reporting System is comprehensive and has been regularly updated over time.

We have added a sentence explaining that all our outcomes are comprehensive and have been regularly updated over time, along with citations supporting the veracity of this statement.

“All hospitals and emergency departments in Ontario report to these databases. DAD, OMERS, and NACRS are updated 4 times per year, and ORGD is updated once per year (21–24).” (p.5) 

6. What is the role of the private sector? Negligible? Should people who are NOT covered by the OHIP pay their medical expenses “out-of-pocket” or are reimbursed by private insurance companies (of course in case there are such companies, as in the US health system).

There are some groups of people not covered by the OHIP in Ontario, but the proportion is small. We added the following clarification on OHIP eligibility:

“Participants must be 18 or over as of October 17, 2018 (the date of cannabis legalization), and have continuous OHIP coverage and residency in Ontario for the entire study period to be included in the study (October 2015 to May 2021). OHIP is Ontario’s universal healthcare that covers over 95% of Ontario residents, those who are not eligible includes individuals in their 3-month waiting periods and migrants with temporary status (e.g. international students, temporary workers)(16), which includes approximately 500,000 people (17).” (p.4-5)

7. The “pre-legalization period (October 2015-October 2018)” (verbatim) cannot be described under the label “Intervention”. Maybe this section could be rather called “Baseline data AND intervention”. There is no intervention from October 2015-October 2018, just more of the same, i.e. the previous status quo.

We changed the section titled “intervention” to “Cannabis legalization phases” (p.5) for accuracy.

8. The statistical analyses are sound. The missing point is to state: we performed two sensitivity analyses and they showed that… WHAT? (e.g. we found “the sufficient convergence of the AR(1) coefficient [was] reached”, quoting https://cran.r-project.org/web/packages/prais/prais.pdf). 

We restructured the sentence to clarify the specific sensitivity tests. Now it reads: 

“We conducted two sensitivity analyses: 1) to examine whether the results were robust regardless of the functional form and address possible residual autocorrelation in a time-series dataset, the Prais-Winsten regressions following a first-order autoregressive process were performed(27), and 2) we included the months March and April 2020 in an alternative specification of the models, and used an indicator variable for both months as an adjustment to confirm whether the estimates were robust.” (p.6) 

9. There is a kind of ecological (?) trend towards an increase of cannabis use over time, even during the period NO intervention has been in place: “In Southern Ontario (Table 3), there was an increasing trend in cannabis-related ED visits across all gender and age groups during the pre-legalization period (verbatim)”. The influence of the actual interventions is clear, but pre-intervention trends should be better discussed. Maybe there is no definitive explanation, but much probably there are some hypotheses: changing attitudes and mores?, a higher use among younger birth cohorts, even in the absence of any legal change?, etc.

We added a line in our discussion section briefly describing a possible mechanism for this trend: 

“Prior to cannabis legalization, we observed an increasing trend in cannabis-related ED visits in all subpopulation groups, which may reflect decreased stigma associated with recreational cannabis-use over time(28).” (p.8)

10. Please, explain how a place described by the authors themselves (in the Introduction) as SO different from other regions shows trends roughly comparable to other Canadian provinces: “Our results are largely consistent with findings from prior research that found cannabis legalization (Phase 1) (6) and cannabis edibles legalization and commercialization (Phase 2) were associated with increases in cannabis-related ED visits in Canadian provinces”

We have now added the following explanation into the discussion section: 

“ It is important to put these seemingly large relative increases into context. The issues associated with Northern Ontario (e.g. higher substance use problems) seems to be reflected in consistently higher rates of ED visits in each subpopulation group in the North; however, the effects of legalization appear to be uniform across northern and southern Ontario. A possible explanation is that Northern Ontario may be protected by the relatively lower density of cannabis retailers (35). However, further research is necessary to investigate the association between retailer availability and cannabis-related ED visits.” (p.10)

11. The COVID pandemic should not be addressed as nothing else but a limitation. “Phase 2 of the study overlapped with the COVID-19 pandemic, so we cannot confidently estimate the independent effects associated with commercialization and edibles.” Of course, it is! the authors are 100% right! But what happened in the region under analysis during this period? Several papers, worldwide show an increase (whereas other studies have shown a decrease or no major change) in the use of different substances in consequence of higher levels of anxiety (among other factors). See, for instance, a recent paper from Canada (https://substanceabusepolicy.biomedcentral.com/articles/10.1186/s13011-022-00441-x). Besides being a limitation in the sense of confusing the findings of the intervention, what do the authors know or hypothesize about what has happened in the region under analysis? As described by the authors themselves, this Region seems to be quite different from other contexts.

We have added the following sentence in the manuscript to contextualize what happened during the pandemic and how it contributes to the regional patterns of legalization effects:

“While our study cannot disentangle the effects of COVID vs. commercialization, we can better contextualize our results by looking at changes in cannabis sales over the COVID period. There is evidence that the COVID-19 pandemic was associated with a 25% increase in monthly cannabis sales in Canada, relative to the counterfactual (constructed from the pre-pandemic trend)(36). More specifically, the pandemic-related increased sales found in the prior study was driven by cannabis retailers, which may explain the increased number of subpopulation groups that had significantly increases in ED visits, particularly among younger adults in Southern Ontario (given that it has higher retailer density compared to the North).” (p.10)

---

## [Editor Report · Decision Letter 1]

29 Sep 2022

Associations between recreational cannabis legalization and cannabis-related emergency department visits by age, gender, and geographic status in Ontario, Canada: an interrupted time series study

PONE-D-22-11906R1

Dear Dr. Chum

We’re pleased to inform you that your manuscript has been judged scientifically suitable for publication and will be formally accepted for publication once it meets all outstanding technical requirements.

Kind regards,

Raphael Mendonça Guimaraes, PhD

Academic Editor

PLOS ONE

---

## [Editor Report · Acceptance letter]

5 Oct 2022

PONE-D-22-11906R1 

Associations between recreational cannabis legalization and cannabis-related emergency department visits by age, gender, and geographic status in Ontario, Canada: an interrupted time series study 

Dear Dr. Chum:

I'm pleased to inform you that your manuscript has been deemed suitable for publication in PLOS ONE. Congratulations! Your manuscript is now with our production department. 

Kind regards, 

on behalf of

Dr. Raphael Mendonça Guimaraes 

Academic Editor

PLOS ONE